# Starvation-Dependent Inhibition of the Hydrocarbon Degrader *Marinobacter* sp. TT1 by a Chemical Dispersant

**Saskia Rughöft** [1] ⓘ**, Anjela L. Vogel** [1] **, Samantha B. Joye** [2] ⓘ**, Tony Gutierrez** [3] ⓘ **and Sara Kleindienst** [1,*] ⓘ

[1] Microbial Ecology, Center for Applied Geosciences, University of Tübingen, 72076 Tübingen, Germany; saskia.rughoeft@uni-tuebingen.de (S.R.); anjela.thon@uni-tuebingen.de (A.L.V.)

[2] Department of Marine Sciences, University of Georgia, Athens, GA 30602-3636, USA; mjoye@uga.edu

[3] Institute of Mechanical, Process and Energy Engineering, School of Engineering and Physical Sciences, Heriot-Watt University, Edinburgh EH14 4AS, UK; tony.gutierrez@hw.ac.uk

* Correspondence: sara.kleindienst@uni-tuebingen.de

**Abstract:** During marine oil spills, chemical dispersants are used routinely to disperse surface slicks, transferring the hydrocarbon constituents of oil into the aqueous phase. Nonetheless, a comprehensive understanding of how dispersants affect natural populations of hydrocarbon-degrading bacteria, particularly under environmentally relevant conditions, is lacking. We investigated the impacts of the dispersant Corexit EC9500A on the marine hydrocarbon degrader *Marinobacter* sp. TT1 when pre-adapted to either low *n*-hexadecane concentrations (starved culture) or high *n*-hexadecane concentrations (well-fed culture). The growth of previously starved cells was inhibited when exposed to the dispersant, as evidenced by 55% lower cell numbers and 30% lower *n*-hexadecane biodegradation efficiency compared to cells grown on *n*-hexadecane alone. Cultures that were well-fed did not exhibit dispersant-induced inhibition of growth or *n*-hexadecane degradation. In addition, fluorescence microscopy revealed amorphous cell aggregate structures when the starved culture was exposed to dispersants, suggesting that Corexit affected the biofilm formation behavior of starved cells. Our findings indicate that (previous) substrate limitation, resembling oligotrophic open ocean conditions, can impact the response and hydrocarbon-degrading activities of oil-degrading organisms when exposed to Corexit, and highlight the need for further work to better understand the implications of environmental stressors on oil biodegradation and microbial community dynamics.

**Keywords:** *Marinobacter*; dispersant; Corexit; hexadecane; oil biodegradation; alkane biodegradation; carbon starvation

## 1. Introduction

During major marine oil spills, chemical dispersants are applied routinely with the aim of breaking up surface slicks and dispersing the oil into the water column. Following the Deepwater Horizon blowout in the Gulf of Mexico on April 20th of 2010, seven million liters of dispersant (Corexit EC9500A and EC9527A) were applied in response to the discharge of an estimated 800 million liters of crude oil into the Gulf ecosystem [1,2]. The impacts of the oil spill on the Gulf's ecosystem and the health of its human inhabitants are well documented [3,4]. However, the impact of chemical dispersant application on native microbial oil-degrading populations is still unclear and conflicting reports have led to disputes over best practices [5].

Some studies have documented enhanced hydrocarbon (HC) degradation in the presence of dispersants [6–8], whereas others have suggested toxic or inhibitive effects on microbial oil

degradation [9–12]. A number of different explanations for these contradictory findings have been proposed, ranging from methodological issues (e.g., dispersant concentrations, types and weathering status of crude oil or HCs used) to microbiological and ecological considerations (e.g., species-/strain-specific dispersant responses and the relevance of the native microbial community composition) [7,11,13–15].

To our knowledge, however, the pre-spill conditions that HC-degrading microorganisms face in oligotrophic (i.e., nutrient-/substrate-limited) ocean waters have not been considered as a factor that can predict their physiological response to oil and chemical dispersant exposure. Most published studies have used either laboratory cultures of HC-degrading bacteria supplemented with high substrate and nutrient concentrations, or natural seawater communities in microcosm experiments generously supplemented with nutrients [7,8]. In addition, substrate and nutrient limitations have been shown to affect hydrophobicity, production of extracellular polymeric substances, and biodegradation potential of HC-degrading isolates [16–19]. Few studies have previously investigated the impact of dispersant exposure in combination with different nutrient concentrations on oil biodegradation, and these studies produced conflicting results [10,20,21]. The effects of dispersants on substrate-limited HC degraders remain largely unexplored.

In this study, the marine hydrocarbon-degrading strain *Marinobacter* sp. TT1 (isolated during the Deepwater Horizon spill [22]) was grown on *n*-hexadecane under either substrate starvation or typical well-fed laboratory conditions before the experiment, the former resembling *in situ* conditions of the open ocean. The genus *Marinobacter* includes several ubiquitous HC-degrading species that respond positively to marine oil spills [8,10,23,24]. Furthermore, some *Marinobacter* spp. have been reported to be inhibited by chemical dispersant exposure [8–10,14]. The aim of this study was to investigate the impact of chemical dispersant exposure on the growth and *n*-hexadecane biodegradation activity of *Marinobacter* sp. TT1 cultures pre-adapted to low or high HC substrate concentrations.

## 2. Materials and Methods

### 2.1. Bacterial Strain and Pre-Adaptation of Cultures

The strain *Marinobacter* sp. TT1 was isolated from a deep-sea plume water sample collected during the active phase of the Deepwater Horizon oil spill using *n*-hexadecane for enrichment [22]. In this study, *Marinobacter* sp. TT1 was cultivated in ONR7a minimal medium [25] supplemented with different carbon substrates and grown (dark, 20 °C, 120 rpm on shaker) in half-filled 20 mL glass headspace vials (pre-baked at 300 °C for 8 h) with autoclaved PTFE-lined crimp lids. For inoculation, 200 μL of the respective pre-cultures were transferred.

The well-fed pre-culture was revived from a previously well-fed glycerol stock (*Marinobacter* sp. TT1 in complex medium without added *n*-hexadecane), transferred into liquid ONR7a minimal medium, and subsequently supplied with 100 mg L$^{-1}$ *n*-hexadecane weekly, providing an almost constant substrate supply. The starved pre-culture was revived from a previously starved glycerol stock (substrate-starved *Marinobacter* sp. TT1 in minimal medium that had received *n*-hexadecane before starvation), transferred into liquid ONR7a medium, and subsequently supplied with 50 mg L$^{-1}$ *n*-hexadecane once every three weeks. Thus, the starved pre-culture was adapted to survive periods with no substrate availability, mimicking the conditions of oligotrophic open ocean waters. Both pre-cultures were then transferred with 100 mg L$^{-1}$ *n*-hexadecane once (three days before the start of the experiment), mimicking a large hydrocarbon pulse similar to an oil spill scenario. All pre-cultures were derived from the same original culture and before the start of the experiment both *Marinobacter* sp. TT1 pre-cultures were confirmed to be the same strain by sequencing of the 16S rRNA gene fragment (100% identity) that was amplified using primer pairs 341f/907r [26].

## 2.2. Experimental Setup

All experimental pre-cultures were grown on *n*-hexadecane without prior exposure to Corexit EC9500A (see above). At the start of the respective experiments, the following four culture conditions were set up for both the starved and the well-fed pre-culture: (i) no added carbon substrate (Control); (ii) 100 mg L$^{-1}$ *n*-hexadecane; (iii) 100 mg L$^{-1}$ *n*-hexadecane and 10 mg L$^{-1}$ Corexit; or (iv) 100 mg L$^{-1}$ Corexit. All treatments were run in triplicate and sampled sacrificially after 0, 2, and 5 days, except for the Control and Corexit setups (i) and (iv), respectively, which were only sampled at the start and at the end of the experiment. For *n*-hexadecane + Corexit treatments (iii), Corexit concentrations were chosen according to the recommended maximum dispersant-to-oil ratio of 1:10 for application of Corexit after oil spills [27], whereas a higher concentration of 100 mg L$^{-1}$ was chosen for the Corexit only treatments (iv) because no additional carbon substrates were supplied. Due to concerns about gas phase losses during sampling, separate triplicates were used for *n*-hexadecane ± Corexit setups (ii) and (iii) to obtain samples for cell counts and hydrocarbon quantification analysis. Additionally, abiotic control setups for hydrocarbon quantification were prepared, containing no inoculum and either (v) 100 mg L$^{-1}$ *n*-hexadecane or (vi) 100 mg L$^{-1}$ *n*-hexadecane and 10 mg L$^{-1}$ Corexit.

## 2.3. Cell Counts

For cell counts, samples were fixed with 1% paraformaldehyde and stored at 4 °C until further processing. To reduce cell aggregate formation that would preclude accurate cell count measurements, 1% (*w/v*) ethylenediaminetetraacetic acid (EDTA) was added to the samples before sonication (20% intensity, 30 s; Sonoplus ultrasonic homogeniser, Bandelin electronic GmbH & Co. KG, Berlin, Germany), a procedure that was optimized for this culture. Samples were then filtered onto Isopore polycarbonate membrane filters (GTTP, 0.2 μm; Millipore) and stained with 4′,6-diamidino-2-phenylindole (DAPI; 1 μg mL$^{-1}$) for 10 min, washed with ddH$_2$O for 5 min, rinsed in ethanol (80%), and then air dried in the dark at room temperature. Filtered sample volumes were adjusted to sample cell densities and diluted with cell-free water to ensure the filtration of 10 mL of liquid for all samples. Membrane filters were embedded using a 1:4 mixture of Vectashield mounting medium (Vector Laboratories, Burlingame, CA, USA) and Citifluor AF2 glycerol solution (EMS Acquisition Corp., Hatfield, Pennsylvania) before the slides were analyzed using fluorescence microscopy (Leica DM 5500 B; Leica Microsystems, Wetzlar, Germany). Images were taken at a magnification of 1000× with a Leica DFC 360 FX camera using the Leica Application Suite Advanced Fluorescence software (2.6.0.766). Cell counts of the images were performed using the "Find Maxima" function (noise tolerance = 7) of the Fiji distribution of ImageJ [28], counting a minimum of 20 images and 600 cells per sample. To assess aggregate morphologies, selected subsamples were additionally processed (i.e., filtered and stained) without the sonication step and imaged as described above.

## 2.4. Hydrocarbon Quantification

For *n*-hexadecane quantification, deuterated *n*-hexadecane (D34, Sigma-Aldrich, St. Louis, MI, USA) was added to the samples as an internal standard (20 mg L$^{-1}$) before extracting the entire vial using 9 mL cyclohexane (purity 99.9 %, Carl ROTH, Karlsruhe, Germany). Vials were shaken at 300 rpm for half an hour, the phases were allowed to separate overnight (20 °C), and then subsamples of the cyclohexane were used to quantify the residual *n*-hexadecane via gas chromatography (Agilent 6890N GC; Agilent Technologies, Santa Clara, CA, USA) coupled with mass spectrometry (Agilent 5973 MS). For separation, a J+W Scientific DB-5MS (30 m length, 0.25 mm ID, 0.25 μm film thickness) capillary column was used. The device was operated in a pulsed splitless mode with a helium flow of 0.8 mL/min. Oven temperature was initiated at 65 °C (4 min), then ramped at 10 °C/min to 220 °C, further ramped at 20 °C/min to 310 °C, and held at this temperature for 5 min.

## 2.5. Data Analysis

To test whether differences in cell numbers or *n*-hexadecane concentrations between the cultures grown with and without Corexit at the same time points (or between the starved and well-fed culture) were statistically significant, ANOVA followed by Tukey post hoc test ($p < 0.05$) for multiple comparisons was used. Previously, it was tested if residuals were normally distributed (using Shapiro–Wilks test, $p > 0.05$) and variances were homogenous (tested with Levene test, $p > 0.05$). The R programming language [29] was used to perform all described tests and produce all presented figures.

## 3. Results and Discussion

### 3.1. Rapid Response by Starved Marinobacter sp. TT1 to a High Hydrocarbon Pulse

The initial cell numbers at the start of the experiment were $1.9 \times 10^6$ cells mL$^{-1}$ (well-fed culture) and $3.8 \times 10^6$ cells mL$^{-1}$ (starved culture). After five days, starved *Marinobacter* sp. TT1 had achieved 58% higher cell numbers compared to the well-fed culture ($4.95 \times 10^8$ vs. $2.85 \times 10^8$ cells mL$^{-1}$, respectively; Figure 1A,B) and degraded significantly more of the *n*-hexadecane (0.89 vs. 33.87 mg *n*-hexadecane L$^{-1}$ remained, respectively; $p = 0.0062$; Figure 1C,D). This more robust response of starved cells to a hydrocarbon pulse aligns with previous observations of rapid feast responses in starved bacteria. Starved aquatic bacteria are known to react immediately by expressing very high uptake rates after receiving a "shock" substrate pulse [30]. Further, a bathypelagic *Marinobacter*, maintained under starvation for 1.6 years, grew rapidly after being pulsed with fresh organic matter [31]. The high relevance of this opportunistic lifestyle for HC-degrading marine bacteria, including *Marinobacter* sp. as a known opportunitroph [32], has been discussed previously [33,34]. Similar substrate-pulse responses have also been described for aromatic HC-degrading bacterial strains [16,17].

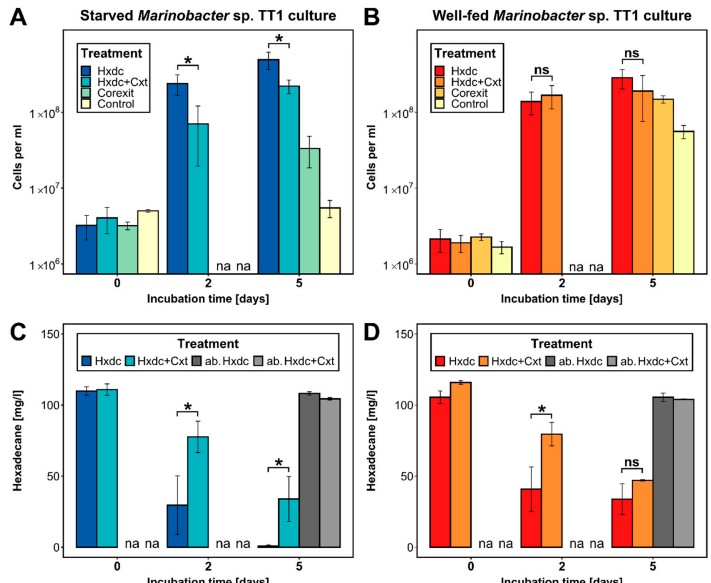

**Figure 1.** Cell numbers (cells mL$^{-1}$) and residual *n*-hexadecane concentrations (mg L$^{-1}$) in cultures inoculated with either starved (**A**,**C**) or well-fed (**B**,**D**) *Marinobacter* sp. TT1 during the incubation period of 5 days (means ± SD; n = 3). *n*-Hexadecane was supplied at 100 mg L$^{-1}$ with or without 10 mg L$^{-1}$ of Corexit. Corexit alone was supplied at 100 mg L$^{-1}$. Abiotic controls were supplied with the same respective concentrations but without inoculum. No carbon source was added to the control treatment. Significance levels are only shown for comparisons of treatments with *n*-hexadecane alone against those with both *n*-hexadecane and Corexit (* $p < 0.05$, ns = not significant). Cell numbers are shown using a logarithmic scale. Hxdc = *n*-hexadecane; Cxt = Corexit; ab. = abiotic control; na = not analyzed.

### 3.2. Only Starved Marinobacter sp. TT1 Was Consistently Inhibited after Corexit Exposure

The growth of the starved culture of *Marinobacter* sp. TT1 was significantly inhibited in the treatment containing both *n*-hexadecane and Corexit when compared to the *n*-hexadecane only treatment. With added Corexit, growth of these cells was 55% lower ($2.22 \times 10^8$ vs. $4.95 \times 10^8$ cells mL$^{-1}$; $p = 0.0103$; Figure 1A) and 30% less *n*-hexadecane was biodegraded (33.94 vs. 0.89 mg *n*-hexadecane L$^{-1}$ remained, $p = 0.0224$; Figure 1C) after five days. In addition, larger cell aggregates were observed in starved cultures with added Corexit (appr. 0.5–1.5 cm long versus <1 mm in *n*-hexadecane only treatments; Figure S1) and fluorescence microscopy revealed a changed, amorphous structure of these aggregates lacking defined, visible cell morphologies (Figure 2A,B). Well-fed cultures, in contrast, showed no significant difference in growth or microscopic aggregate structure between *n*-hexadecane treatments with and without Corexit (Figures 1B and 2D,E), and a smaller difference between aggregate sizes (ranging between 1–5 mm size; Figure S1). Significantly less *n*-hexadecane was degraded in the treatment with both *n*-hexadecane and Corexit compared to the *n*-hexadecane treatment after two days (79.56 vs. 40.93 mg *n*-hexadecane L$^{-1}$ remained, $p = 0.0193$; Figure 1D). However, similar residual *n*-hexadecane was determined in both treatments after five days in well-fed cultures. Interestingly, in the Corexit-only treatments (i.e., in the absence of *n*-hexadecane), growth was observed for both starved and well-fed cultures after five days (from $2.4 \times 10^6$ to $3.3 \times 10^7$ or $1.5 \times 10^8$ cells mL$^{-1}$, respectively; Figure 1A,C), suggesting that *Marinobacter* sp. TT1 can use Corexit components as a growth substrate. No macroscopic cell aggregates were observed. Notably, the well-fed culture reached significantly higher cell numbers under these conditions ($p = 0.0042$). This could be explained by a more pronounced inhibition of the starved culture caused by Corexit exposure or by a longer lag phase in the starved culture when adapting to a new substrate. Growth on Corexit compounds has previously been reported for other HC degraders, such as *Colwellia* sp., *Alcanivorax* sp., and *Acinetobacter* sp. [13,35].

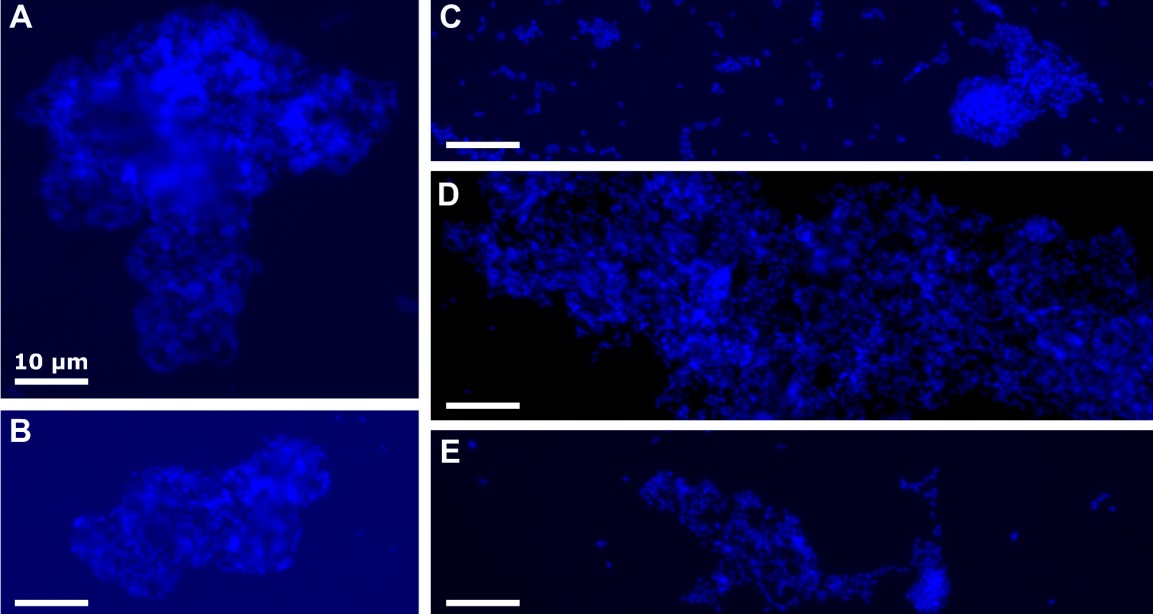

**Figure 2.** Fluorescence microscopy images of aggregates in *Marinobacter* sp. TT1 cultures after five days of incubation. Cells are stained with 4',6-diamidino-2-phenylindole (DAPI; appears blue). (**A,B**) Amorphous aggregates from starved cultures that received both 100 mg L$^{-1}$ *n*-hexadecane and 10 mg L$^{-1}$ Corexit. (**C**) Aggregates from starved cultures that received only 100 mg L$^{-1}$ *n*-hexadecane (i.e., without Corexit). (**D,E**) Aggregates from well-fed cultures that received only 100 mg L$^{-1}$ *n*-hexadecane (**D**) or both 100 mg L$^{-1}$ *n*-hexadecane and 10 mg L$^{-1}$ Corexit (**E**).

### 3.3. Potential Mechanisms Responsible for the Observed Effects of Starved Marinobacter sp. TT1

The underlying mechanisms of the pronounced response of starved *Marinobacter* sp. TT1 to dispersant exposure still remain to be elucidated. However, a number of recognized starvation effects could trigger the observed inhibition effects. First, changes in cell surface hydrophobicity, cell motility, and production of extracellular polymeric substances occur in response to starvation in marine bacteria [18,19,36,37] and these cell properties are important for a successful HC-degrading lifestyle [38]. Based on our observations, a modified production of extracellular polymeric substances induced by dispersant exposure in starved cells might have led to the amorphous cell aggregate structure at the microscopic level. Second, carbon starvation responses can induce membrane modifications, such as the degradation of membrane phospholipids [39–41], which could lead to higher membrane sensitivity to dispersant components (e.g., surfactants and solvents) and, in turn, compromise cell physiologies. Membrane modifications could also be another reason for the observed amorphous aggregate structures. Additionally, key proteins in alkane metabolism (i.e., AlkB, AupA/B) are associated with the cell membrane [42,43] and could thus be detrimentally affected in starved cells exposed to chemical dispersants.

The ecological theory of multiple stressors can also explain these results. Organisms are often exposed to multiple stressors at once under in situ conditions and the cumulative biological effects of these stressors can be synergistic, i.e., distinct from the additive effects of individual stressors [44,45]. Synergistic effects of dispersants and other stressors have been described in a few studies. For example, Corexit 9257 inhibited oil biodegradation the most under nutrient limited conditions [20]. Similarly, phytoplankton was more sensitive to dispersants under nutrient-limited conditions [46] and the HC degrader *Rhodococcus* sp. PC20 was more inhibited by dispersant exposure under high pressure conditions [12]. Therefore, (previous) environmental stress—in this case substrate limitation—can lead to different microbial responses to dispersant or dispersant-hydrocarbon exposure, potentially explaining the observed, more severe response of starved hydrocarbon degraders to chemical dispersants.

### 3.4. Environmental Implications

Although some previous studies reported clear inhibition responses of *Marinobacter* when exposed to chemical dispersants [9,10], other experiments revealed both inhibited and stimulated members of the genus *Marinobacter* [8,14,15]. Ecotype-specific dispersant responses might explain some of these previous observations (as discussed in, e.g., [14]), but this study is the first to identify the adaptation to low substrate availability as a factor that affects how marine HC-degrading microorganisms could respond during an oil spill at sea when chemical dispersants are applied. Due to the inherent spatial and temporal heterogeneity of substrate and nutrient availability in the marine environment, the observations reported here could have wide reaching environmental implications. Contrary to obligate HC-degrading bacteria, members of *Marinobacter* are known as ubiquitous, opportunistic heterotrophs that can use a wide range of substrates [23,24]. However, their distribution in the marine environment is largely confined to oligotrophic (i.e., nutrient-/substrate-limited) waters which make up at least 18% (estimated oligotrophic gyre area [47,48]) of the global ocean. Consequently, *Marinobacter* spp. are adapted to surviving long periods of low substrate availability [31,32,49]. The substrate history of marine HC degraders depends on their lifestyle (e.g., free-living or particle-attached), their niches (e.g., surface or deep waters, polar or tropical latitudes), the dynamics of their environment (e.g., local seasonality, bloom regimes), and the levels of natural and anthropogenic substrate emissions (e.g., chronic or dynamic HC inputs via natural oil seeps, ship traffic, or accidental oil releases). A few studies have also described physiological adaptations to substrate limitations in *M. hydrocarbonoclasticus* SP17 by comparing sessile cells growing in *n*-hexadecane-associated biofilms and the planktonic cells released from the biofilm which need to survive for an indefinite period without a new substrate source in the environment [49,50]. These reports provide further evidence for the environmental relevance of previously experienced substrate limitation in marine HC-degrading

bacteria and the importance of considering this factor when assessing impacts of stressors like chemical dispersants. According to the multiple stressor theory, additional stressors such as nutrient limitation and non-HC pollutant exposure (e.g., heavy metals, pesticides, halogenated compounds) probably also play a role in modulating the sensitivity of oil-degrading microorganisms to chemical dispersant exposure in the marine environment. Based on the presented findings, all of these factors could play a role in determining the impacts of chemical dispersants on marine in situ HC biodegradation. Additionally, our results suggest that chemical dispersant exposure affected extracellular polymeric substance production and/or aggregate formation in starved *Marinobacter* sp. TT1. Similar dispersant impacts were further confirmed in a recent comparative proteomics study performed in our lab [51]. Observations made during the Deepwater Horizon spill in relation to marine oil snow formation and sedimentation suggest that these dispersant-induced effects are environmentally relevant and could determine the fate of spilled oil in the marine environment [52,53].

## 4. Conclusions

To our knowledge, this is the first study demonstrating that the pre-adapted state (i.e., substrate history) of marine hydrocarbon-degrading bacteria can have a significant effect on how these organisms respond to chemical dispersant use in the event of an oil spill. These observations help to explain previously inconsistent findings about dispersant impacts in the literature and highlight the need for considering in situ environmental conditions (e.g., oligotrophic versus copiotrophic substrate/nutrient adaptations) when conducting laboratory experiments in which a single change in parameters could propagate and lead to different findings, e.g., underestimated dispersant impacts. To better inform future decisions on dispersant use in marine oil spill situations, there is a critical need for additional baseline knowledge of environmental microbial communities, including their nutritional status, and for systematic screenings of cultured representatives regarding their response to dispersants under environmentally relevant conditions, as well as more insights into how and why dispersants might affect their physiology.

**Supplementary Materials:** The following are available online at http://www.mdpi.com/2077-1312/8/11/925/s1, Figure S1: Aggregate morphology in *Marinobacter* sp. TT1 cultures after five days of incubation.

**Author Contributions:** Conceptualization, S.K., S.B.J. and S.R.; investigation, S.R. and A.L.V.; resources, S.K. and T.G.; data curation, S.R.; writing—original draft preparation, S.R. (major part) with S.K.; writing—review and editing, S.B.J., T.G., A.L.V.; visualization, S.R.; supervision, S.K.; funding acquisition, S.K. All authors have read and agreed to the published version of the manuscript.

**Funding:** This study was funded by the Baden-Württemberg Foundation's Elite Program for Postdocs and by the Deutsche Forschungsgemeinschaft (DFG, German Research Foundation—fellowship grant #326028733). We acknowledge support by the Open Access Publishing Fund of the University of Tübingen.

**Acknowledgments:** The authors would like to thank the U.S. National Oceanic and Atmospheric Administration for providing Corexit EC9500A, Renate Seelig and Peter Grathwohl for GC-MS measurements and Anja Pohl for help with culture maintenance.

**Conflicts of Interest:** The authors declare no conflict of interest.

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
