# Peer review of "Starvation-Dependent Inhibition of the Hydrocarbon Degrader Marinobacter sp. TT1 by a Chemical Dispersant"

_jmse, doi:10.3390/jmse8110925_

Round 1
Reviewer 1 Report
Dear Authors,
The Short Communication “Starvation-dependent inhibition of the hydrocarbon degrader Marinobacter sp. TT1 by a chemical dispersant” submitted to the Journal of the Marine Science and Engineering by Rughöft et al., highlights the cap in our knowledge about the effect of the chemicals used during the oil spill recovery on indigenous marine microbial populations.
Comments:
“marine hydrocarbon degrading (Marino)bacter” or “marine hydrocarbon-degrading (Marino)bacter” is there a difference? Please check through the manuscript!
Heading of the manuscript is not clear and requires improvement.
Graphical abstract - lower part (well-fed culture) is not in the accordance with the main text!
Materials and Methods
How many biological and technical parallels did you have?
Line 93 How was chosen the concentration of the Corexit? Why was the concentration of Corexit 10 times higher when supplemented as single substrate?
What was the initial cell number in the experiments?
Lines 100-113 –
How big was the volume of the sample that was filtered? What was the membrane filter size?
Why did you use DAPI for counting the cells? The easiest way could be the counting chamber. Or if you had problems with aggregations you could use some DNA-based method.
Why did you use sonication (30 seconds!)? It will disrupt the cells and as DAPI binds to the DNA you can’t determine cell morphology at all!
Lines 114-116 Aggregates were observed in samples that were not sonicated, but did you use filtering? Please specify.
Results and discussion
Lines 136-139 You are comparing averages, but SD are quite big. Is the difference statistically significant? If you compare the results of the two cell-lines did you got statistically significant differences?
Physiological state of the starved and well-fed cells is different, starved cells getting “good food” will use it faster than the well-fed that are getting it regularly.
Fig 1A and B – Please use logarithmic values on the y-axis. The scale (0-point) of the y-axis is not correct!
Fig 1B – How can the cells grow if no substrate is added to the growth medium?
Fig 2 The aggregates of the well-fed cells on the Fig 2D are longer than the starved cells.
Lines 163-164 “fluorescence microscopy revealed a changed, amorphous structure of these aggregates lacking defined, visible cell morphologies (Figure 2A, B).“ - Please provide “normal cell” images. If these aggregates were 0,5-1,5 cm long then under the microscope they are also big and you can’t get clear images as there are so many layers of the cells. Why were the aggregates of the starved cells grown on hexadecane smaller than well-fed cell aggregates on same substrate (Fig S1)?
Lines 167-170 “Significantly more n-hexadecane was degraded in the treatment with both n-hexadecane and Corexit compared to the n-hexadecane treatment after two days (79.56 vs. 40.93 mg n-hexadecane l-1 remained, p = 0.0193; Figure 1D).“ -On the provided Fig 1D opposite results can be observed. Clarify!
Line 170 Modify ”….residual n-hexadecane concentration was determined…”
Lines 174-176 „Notably, the well-fed culture reached higher cell numbers under these conditions, which is likely explained by the more pronounced inhibition effect of Corexit exposure on the starved culture.“ - Providing “foreign” substrates (Corexit) the starved cells need more time to activate specific enzyme systems than well-fed ones.
Have you considered the role of the oxygen limitation in the case of your experiments?
To conclude, the subject is interesting but current manuscript is not ready for the publication as several points must be first corrected. Also, the conducted experiments give us a clue that the effect of the Corexit is different to the starved cells, but no experiments were done to clarify the observations (last two paragraphs of the Results and discussion contain only hypothesis and discussions). The Short Communication format is usually important for the publication of the time-sensitive results as they are relatively short (for example, those in highly competitive or fast-changing disciplines). Is the current paper time-sensitive? I propose that data presented here can be included to the unpublished manuscript referred here as reference 50 (Rughöft, S.; Jehmlich, N.; Gutierrez, T.; Kleindienst, S. Comparative proteomics of Marinobacter sp. TT1 reveals Corexit impacts on hydrocarbon metabolism, chemotactic motility and biofilm formation. Unpublished).
Reviewer 2 Report
This ms describes the effect of n-hexadecane (mimicking oil-spill) and n-hexadecane plus a dispersant (mimicking oil-spill treatment) on a starved and non-starved (testing oligotrophic conditions in the ocean). The ms is well written and targets import albeit poorly studied issue, i.e. the nutrient-stress conditions of bacteria when oil spill occurs. Overall, the amount of data presented would rather fit a comment or note than a full paper, i.e. the discussion/implications seems to far-reaching based on the data sate
Specific comment:
How long were the samples stored for epifluorescence microscopy before analysis? It is well know that bacteria can degrade rapidly at °4C even at the used preservation conditions. This could influence the detected cell numbers.
Reviewer 3 Report
Overview
The question investigated is an important one, and the experiments were generally well designed and presented. However, I have on big question about the interpretation – see comment on line 167.
Specific questions
77 Had the glycerol stocks been prepared from cultures grown on n-hexadecane as sole carbon source? Presumably, but should be stated.
167 “Significantly more n-hexadecane was degraded in the treatment with both n-hexadecane and Corexit…”: From Figs. 1D and 1C, it would appear to be the opposite – the red bar (Hxdc) is lower than the orange bar (Hxdc + Cxt), and the blue bar is lower than the teal one. Is something labeled backwards? How does this affect the rest of the text?
170 “similar residual n-hexadecane was noted in both treatments after five days”: True for the well-fed culture, but not the starved one: should be reworded to make this clear.
173 Maybe better “suggesting” than “indicating”, it is hard to absolutely rule out the presence of some other carbon source at low levels.
Fig. 2 Perhaps a minority opinion, but aggregates would be easier to see in black and white than in color.
Fig. S1. Were vials shaken before pictures were taken, or is this their usual appearance? This could of course affect apparent aggregate size.
General questions
How did growth rates and per-cell degradation rates compare for the starved and well-fed cultures? Per-culture rates may be more comparable to environmental ones, but per-cell rates might shed some light on physiological aspects. They are alluded to in places, but could be shown as a table or figure.
What other carbon sources (that might be found in seawater) is TT1 known to grow on?
English suggestions
125 “helium” (no capitalization)
222 “their ubiquity is confined” is an oxymoron – better something like “Marinobacter are ubiquitous in oligotrophic waters…” or “While globally distributed, Marinobacter are largely confined to …”
Round 2
Reviewer 1 Report
Reviewer 3:
Comments:
Heading of the manuscript is not clear and requires improvement.
Response: We do not identify any issue with the title and neither have the other reviewers, but we welcome the reviewer to explain what specific concern they appear to have with it.
Starvation-dependent inhibition of the hydrocarbon degrader Marinobacter sp. TT1 by a chemical dispersant
The Title is promising more than the tests performed. Only hexadecane was used in experiments, so it should be used instead of hydrocarbon. It is difficult to understand what exactly was inhibited. I would suggest the consideration of the following Titles “Starvation-dependent inhibition of the degradation of hexadecane in Marinobacter sp. TT1 by a chemical dispersant” or ”Chemical dispersant inhibits the degradation of hexadecane by starved Marinobacter sp TT1” .
Graphical abstract - lower part (well-fed culture) is not in the accordance with the main text!
Response: The graphical abstract illustrates the observed results in a simplified manner, showing that the starved culture showed inhibition in growth and degradation, whereas the well-fed culture did not. If the reviewer is referring to some issue of inconsistency with respect to terminology, we have double-checked this.
Here you compare two cell-lines, but in the text only differences inside the one cell-line is discussed! In the Fig 1 the cell numbers and hexadecane degradation levels of the two cell lines are not so drastically different as they are shown in Graphical abstract.
Materials and Methods
How many biological and technical parallels did you have?
Response: Every experimental treatment was run in triplicate, i.e. 3 biological replicates were used in all instances and which is indicated in the text (see L. 93-94).
Half of the question is not answered! How many technical parallels did you have? For example, determination of the concentrations of hexadecane – how many times was one biological replicate analysed by GC-MS?
What was the initial cell number in the experiments?
Response: The initial cell numbers at the start of the experiment were 1.9 x 106 cells per ml (well-fed culture) and 3.8 x 106 cells per ml (starved culture). This information was also added to the paper (lines 140/141).
Based on the Fig 1. it is approximately 0! Correct the figure! NB! Y-axis can start also from 1x106!
Lines 100-113 –
How big was the volume of the sample that was filtered? What was the membrane filter size?
Response: The filtered volume of each triplicate of samples was adjusted according to the cell density in different samples and ranged between 0.02 – 1 ml. The membrane filters had a diameter of 25 mm.
0.02 ml filtered through 25 mm filter??? Really? This volume is not covering the filter, your sample is not spread over the filter equally! How did you calculate cell number/ml?
Why did you use DAPI for counting the cells? The easiest way could be the counting chamber. Or if you had problems with aggregations you could use some DNA-based method.
Response: A counting chamber is commonly used to count eukaryotic (larger-sized) cells (e.g. micro-algae and animal cells), but not bacteria as they are too small to distinguish other non-bacterial small particles. Staining and counting bacteria using DAPI is an established method, which is straight forward. It is additionally useful to directly visualize bacterial cells associated with aggregates, as has been used in many other papers.
Counting chamber is used for the determination of number of bacterial cells! In your case you are using pure culture and quite big cells. Also, you wrote that aggregates were destroyed with sonication. These are the main problems that may raise with the analysis of the environmental samples.
Question from Reviewer 1
How long were the samples stored for epifluorescence microscopy before analysis? It is well know that bacteria can degrade rapidly at °4C even at the used preservation conditions. This could influence the detected cell numbers.
Response: The PFA-fixed samples were stored for 3-5 days (well-fed culture) or 2 months (starved culture) at 4°C before staining and analysing them, with the difference between cultures due to the fact that the respective experiments were performed separately and after one another. While some cell loss could have occurred during this storage time, this would have affected all samples from the respective pre-culture in a similar way and would not have affected the comparison of Hxdc and Hxdc+Cxt treatments. One could imagine this would have impacted the starved culture samples the most, but we observed higher cell numbers in the Hxdc samples from this culture than in all well-fed culture samples. We do not believe that this issue affected our data in a significant way.
Why did you use so different storage times? How does it affect the aggregation? Maybe this is the one reason why you saw bigger aggregates in starved culture?
Results and discussion
Lines 136-139 You are comparing averages, but SD are quite big. Is the difference statistically significant? If you compare the results of the two cell-lines did you got statistically significant differences?
Response: We tested the differences in data from Hxdc and Hxdc+Cxt treatments for statistical significance (see also ‘Data analysis’), and we indicated in the main text (L. 165-167, 175) and in Figure 1 the differences that were statistically significant. We only compared the results of different treatments within either the starved or the well-fed culture.
Why did you make the comparisons only within one culture? For the reader it would be interesting to know did you saw differences also between two cell-lines.
Fig 1A and B – Please use logarithmic values on the y-axis. The scale (0-point) of the y-axis is not correct!
Response: Using a logarithmic scale would obscure the differences between treatments so we used a linear scale and elect to keep this scale.
If there was difference in linear scale it will be also on logarithmic scale and initial cell number of the samples will be presented correctly. Correct the Fig 1.
Fig 2 The aggregates of the well-fed cells on the Fig 2D are longer than the starved cells.
Response: The aggregates shown in the pictures only represent a few examples regarding the observed sizes and lengths. The main difference between starved culture aggregates in Hxdc+Cxt treatments (inhibited treatments) and all other aggregates was with respect to the structure of these aggregates.
Look previous comments.
Lines 163-164 “fluorescence microscopy revealed a changed, amorphous structure of these aggregates lacking defined, visible cell morphologies (Figure 2A, B).“ - Please provide “normal cell” images. If these aggregates were 0,5-1,5 cm long then under the microscope they are also big and you can’t get clear images as there are so many layers of the cells. Why were the aggregates of the starved cells grown on hexadecane smaller than well-fed cell aggregates on same substrate (Fig S1)?
Response: We assume that the reviewer is referring to images of cells that were used for cell counts, so we have attached a representative picture below. However, we underscore the difference between the macroscopically visible aggregates shown in Figure S1 and the microscopic images of smaller aggregates in the same cultures shown in Figure 2. We do not believe that the differences in microscopic aggregate structure (L.169-170) are caused by aggregates being too big or ‘deep’ and thus making clear images impossible. The aggregates shown in Figure 2A and Figure 2D, for example, are both clearly composed of several layers of cells, but while the cells in the 2D aggregate are clearly defined, a more undefined, amorphous structure was observed in the 2A aggregate and this difference was systematically observed between the starved culture Hxdc+Cxt treatment and all other treatments, as described above.
Look previous comments.
Have you considered the role of the oxygen limitation in the case of your experiments?
Response: Since the culture vials we used were only half-filled, we do not believe that oxygen limitation significantly affected our experiments. Assuming that about 12 ml of headspace and 10 ml of liquid medium were in each vial, a total of about 114 µmol of O2 was available per vial, while a maximum of 108 µmol of O2 would have been needed to completely oxidize all available n-hexadecane (100 mg l-1 in 10 ml equals 4.42 µmol) to CO2 and H2O. Since substantial microbial growth was also observed in all treatments with added nhexadecane, we can assume that less O2 than the calculated maximum was consumed and therefore, strain TT1 was likely not oxygen limited in our experiments.
Oxygen is not needed only for degradation of the hexadecane (exogenous oxygen consumption) but also for other metabolic processes (endogenous oxygen consumption). Does the strain TT1 form aggregates also in other media and growth conditions?
To conclude, the subject is interesting but current manuscript is not ready for the publication as several points must be first corrected. Also, the conducted experiments give us a clue that the effect of the Corexit is different to the starved cells, but no experiments were done to clarify the observations (last two paragraphs of the Results and discussion contain only hypothesis and discussions). The Short Communication format is usually important for the publication of the time-sensitive results as they are relatively short (for example, those in highly competitive or fast-changing disciplines). Is the current paper time-sensitive? I propose that data presented here can be included to the unpublished manuscript referred here as reference 50 (Rughöft, S.; Jehmlich, N.; Gutierrez, T.; Kleindienst, S. Comparative proteomics of Marinobacter sp. TT1 reveals Corexit impacts on hydrocarbon metabolism, chemotactic motility and biofilm formation. Unpublished).
Response: We believe that our responses and clarifications have improved the paper. We believe that our findings are highly relevant and of interest to the readership of JMSE as a short communication that will undoubtedly inspire further investigations of this newly identified phenomenon, i.e. previously starved hydrocarbon degraders reacting more sensitively to a chemical dispersant. Studies investigating the effects of synthetic chemical dispersants are rare, and in the available literature, their conclusions are contentious, partly because no report has, hitherto, examined the direct effects of synthetic dispersants on cell growth and hydrocarbon degradation. Our study here has done this and, thus, we believe strongly that these data warrant a stand-alone publication.
I agree that study is interesting and relevant, but in my opinion, there are still some points that should be clarified/corrected before the publication of the manuscript.
Reviewer 2 Report
All comments sufficiently answered.
Author Response
Thank you very much for your suggestions and for agreeing with our revision!